# Deuterium Retention and Release Behavior from Beryllium Co-Deposited Layers at Distinct Ar/D Ratio

**Paul Dinca** [1], **Cornel Staicu** [1], **Corneliu Porosnicu** [1,*], **Oana G. Pompilian** [1], **Ana-Maria Banici** [1], **Bogdan Butoi** [1], **Cristian P. Lungu** [1] and **Ion Burducea** [2]

1. National Institute for Laser, Plasma and Radiation Physics, 409 Atomistilor Street, 077125 Magurele, Romania; paul.dinca@inflpr.ro (P.D.); cornel.staicu@inflpr.ro (C.S.); oana.pompilian@inflpr.ro (O.G.P.); ana.niculescu@inflpr.ro (A.-M.B.); bogdan.butoi@inflpr.ro (B.B.); cristian.lungu@inflpr.ro (C.P.L.)
2. Horia Hulubei National Institute of Physics and Nuclear Engineering, 077125 Magurele, Romania; bion@nipne.ro
* Correspondence: corneliu.porosnicu@inflpr.ro

**Abstract:** Beryllium-deuterium co-deposited layers were obtained using DC magnetron sputtering technique by varying the $Ar/D_2$ gas mixture composition (10/1; 5/1; 2/1 and 1:1) at a constant deposition rate of 0.06 nm/s, 343 K substrate temperature and 2 Pa gas pressure. The surface morphology of the layers was analyzed using Scanning Electron Microscopy and the layer crystalline structure was analyzed by X-ray diffraction. Rutherford backscattering spectrometry was employed to determine the chemical composition of the layers. D trapping states and inventory quantification were performed using thermal desorption spectroscopy. The morphology of the layers is not influenced by the $Ar/D_2$ gas mixture composition but by the substrate type and roughness. The increase of the $D_2$ content during the deposition leads to the deposition of Be-D amorphous layers and also reduces the layer thickness by decreasing the sputtering yield due to the poisoning of the Be target. The D retention in the layers is dominated by the D trapping in low activation binding states and the increase of $D_2$ flow during deposition leads to a significant build-up of deuterium in these states. Increase of deuterium flow during deposition consequently leads to an increase of D retention in the beryllium layers up to 300%. The resulted Be-D layers release the majority of their D (above 99.99%) at temperatures lower than 700 K.

**Keywords:** Be-D layers; deuterium desorption; D flow; D retention

## 1. Introduction

The high paced growth of the human civilization in the last century is strongly tied to the current demand on energy supply. During this time, the overwhelming majority of the required energy was produced by burning fossil fuels, and well into the 21st century it still represents the main engine that powers the automotive and industrial sectors. However, increasing global energy demands in developing countries coupled with the large emission of greenhouse gases released by fossil fuels raise serious concerns on climate change and environmental pollution. To solve these issues, it is required to accelerate the development of clean and sustainable energy sources, primarily as an alternative for the dependence on fossil fuels [1,2]. Renewable energy that harnesses the power of wind and sun can have a major impact in covering the energy demand especially for domestic consumption, however the discontinuity in energy production and heavy reliance on external meteorological factors and geography means that in most cases these energy sources will be complementary to conventional ones.

In this context, nuclear fusion represents an attractive alternative that promises to deliver massive amounts of clean energy to cover this ever-increasing demand. Currently, the technology relies on the fusion of light hydrogen isotopes, deuterium (D) and tritium (T), in a magnetically confined device (tokamak). Several thermonuclear fusion reactors

were built for experimental purposes, among them one of the most important is Joint European Torus (JET) which also represents a test platform for the development of the International Thermonuclear Experimental Reactor (ITER), considered the biggest scientific and technological achievement in the nuclear fusion field.

Since the beginning of the Engineering Design Activities (EDA) phase of the ITER project, a large number of problems concerning the fusion plasma, magnetic confinement, the choice of divertor and first wall materials have been addressed; however, despite this sustained effort, there are still many issues that need to be clarified. An important field of research concerning the design of a fusion reactor is represented by the plasma–wall interaction (PWI). Among the repercussions of PWI, erosion, migration and re-deposition of materials that make up the plasma facing components (PFC), can have the most dramatic consequences on a fusion reactor life-time. One of the consequences of the PFC erosion is that materials can migrate into the fusion plasma resulting in yield loss due to energy absorption and plasma instabilities (edge localized modes) that can damage the reactor. Another unwanted consequence is the co-deposition of the eroded materials on other components of the fusion reactor. The thin layers resulted in most cases exhibit different structural, morphological and chemical properties than the PFC surfaces onto which are deposited, in the process altering their original intended designation [3,4].

ITER safety regulation is meant to maintain the in-vessel retention below the imposed limit of 700 g of T. If this limit is reached, the shutdown of the reactor will be mandatory [5]. Erosion and re-deposition will further complicate this issue giving rise to increased levels of T retention. ITER vessel design consists of two PFC materials, namely, beryllium (Be) and tungsten (W), Be tiles will compose the first wall (700 $m^2$) and W will be employed for the divertor area (50 $m^2$) [6,7]. These materials were selected during the early EFDA phase of the ITER project due to their advantageous plasma compatible properties [8]. Be was considered an excellent candidate as a PFC for ITER due to its high thermal conductivity, low Z, good thermo-mechanical properties and high melting point (1560 K) [9,10]. Be was first tested in a fusion environment during the early stages of the JET project, where it was used as a plasma limiter. The results showed that its use led to a significant reduction of plasma instabilities. Later, in 2008, it was used in JET to replace the graphite PFC in the first wall area of the vessel to test the ITER blanket [11]. Compared to graphite, the use of Be PFC proved to be crucial, leading to a reduction of the oxygen levels inside the reactor. Furthermore, the absence of chemical sputtering leads to a reduction of H isotope retention in comparison to graphite. At the same time, it was showed that Be presence in fusion plasma does not influence significantly the plasma performance [12,13]. However, despite its plasma compatible properties, Be has an increased sputtering yield when subjected to intense D and T particle bombardment and this will consequently lead to the erosion, migration and re-deposition of Be layers. Co-deposition of Be together with nuclear fuel underlines the T retention problem as co-deposited layers will exhibit different retention and release properties [14–16]. Furthermore, it is expected that Be co-deposited layers will be the main contributor to T retention in ITER [17,18].

Considering the above mentioned, it is important to study the H-isotope retention in Be layers in the laboratory frame using D as a surrogate for T [19]. Additionally, it is of the outmost importance to select a suitable deposition technique to obtain similar layers with the ones that can occur in a fusion reactor. It was shown that Be co-deposited layers in JET exhibit the same columnar growth as their counterparts obtained using PVD techniques [20]. Another resemblance with PVD in terms of morphology and grain orientation was observed in PISCES-B linear plasma device for Be films grown under ITER relevant conditions [21]. Taking into consideration all of the above and the pressure inside ITER during operations (0.01–30 Pa), one of the suitable techniques to reproduce these Be layers in laboratory conditions is direct current magnetron sputtering. The main aim of this study was to investigate the deuterium retention and thermal release from Be-D layers, co-deposited at various D flows. To assess only the influence of this parameter, the substrate temperature, gas mixture pressure and deposition rate were maintained stable

throughout the process. A total of four batches of samples were obtained by varying the flow of D into the coating chamber. This study can provide useful information regarding the influence of the conditions in ITER on the morphology, structure and deuterium retention properties of the Be re-deposited layers. Based on the study of the D thermal release behavior, the utility of the tritium removal procedure in ITER can be assessed for the Be-D layers. The structure, morphology and composition of the resulted samples were analyzed using scanning electron microscopy (SEM), X-ray diffraction and Rutherford backscattering spectrometry (RBS). Deuterium release kinetics were investigated using thermal desorption spectrometry (TDS).

## 2. Materials and Methods

### 2.1. Layer Deposition

Beryllium and deuterium-containing layers were grown using direct current magnetron sputtering method. The deposition was performed in a cylinder chamber shape with a 0.05 m$^3$ volume (400 mm diameter and 400 mm height) provided with vacuum system composed of a turbo molecular pump (Turbo-V 750/850-AG, Agilent Technologies, Craven Arms, UK) backed by a dry scroll pump (Bluffton Motor Works, Bluffton, IN, USA). Prior to the sputtering process the deposition chamber was evacuated down to a base pressure of 10$^{-6}$ mbar. The experimental setup is illustrated in Figure 1.

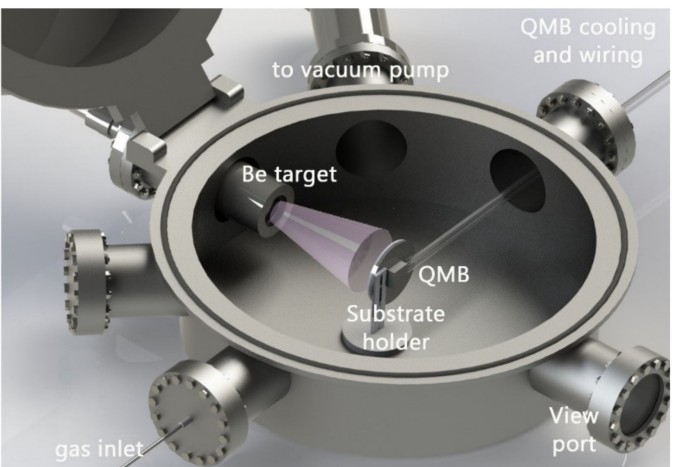

**Figure 1.** Schematic representation of the deposition setup.

Be-D layers were deposited on three types of substrates, namely rectangular tungsten (99.95%), silicon (99.999%) and graphite (99.95%) with 12 mm × 15 mm size. The substrates were ultrasonically cleaned (GTS, Greentechno, Heras, Spain) for 15 min in a mixture of acetone and isopropyl alcohol in order to remove the debris resulted from the polishing process. Subsequently, the samples were washed abundantly using distilled water and left to dry in atmospheric air. The final step consisted in mounting the substrates onto a circular holder placed inside the vacuum chamber at a distance of 10 cm against the sputtering source.

The sputtering source consisted in a water-cooled magnetron cathode provided with a high-purity (99.95%) circular beryllium target (3 mm thickness with 50 mm in diameter). Beryllium cathode was operated in DC using a 500 mA current and 350 V voltage, respectively. These parameters were used for all depositions. Before starting the deposition process, the target was pre-sputtered for 30 min in pure Ar (99.9999%) gas at a pressure of 1 Pa, to eliminate the BeO layer formed at the surface and other retained impurities from the exposure to atmospheric air. In this stage, the substrates were shielded from the sputtered material using a shutter located between the target and the sample holder. After this step D gas (99.996%) is fed into the vacuum chamber in a controlled manner using a mass flow controller (MFC) Alicat Scientific (Tucson, AZ, USA) type. The Be-D

layers (four in total) were obtained by adjusting the D/Ar mass flow ratio between 0.1 and 1 using a fixed Ar flow of 20 sccm and adjustable D flow (2 sccm–20 sccm). The total pressure resulted from summing the partial pressure of argon and deuterium was kept constant (2 Pa) for all performed depositions, by adjusting manually the aperture between the vacuum chamber and the turbo molecular pump. During the coating process a $-40$ V bias voltage was applied on the substrate with the purpose to enhance the ion bombardment on the growing layer. The current density measurements were performed before deposition. The substrate holder was replaced by a planar probe with the same dimension (10 cm diameter) and placed in a virtual position identical to the samples. The results obtained in similar conditions in which coatings were obtained showed that the current density changed between 0.25–0.3 mA/cm$^2$. The current density on the substrates is influenced by the power applied on the cathode. The increase of the input power leads to a higher electron density which in turn can facilitate the ionization processes. However, in our study the input power was stable and the current density changed with the gas mixture composition. For both D2 and D4 the current density was 0.3 mA/cm$^2$ and decreased to 0.27 mA/cm$^2$ for D10 and 0.25 mA/cm$^2$ for D20, respectively. However, extensive measurements are required to clearly underline that the decreasing trend is linked to gas composition and are beyond the scope of our study.

During deposition the samples are heated by the magnetron plasma up to a temperature of 343 K. For a better control of the final aimed thickness of 500 nm, the deposition rate was measured in situ using a quartz microbalance (INFICON, Bad Ragaz, Switzerland) and it was maintained at 0.06 nm/s. In our current experimental setup, the quart microbalance represents an integrated component of the holder assembly. The quartz microbalance has a fixed position in the center of the holder with the substrates arranged around it. It allows the recording of the deposition rate throughout the process and can help to fine-tune the desired thickness of the layers. It was observed by Temmerman et al. [20] that deposition rate and temperature are fundamental parameters that affect the deuterium retention and inventory in Be-D layers, as the current contribution is focused on the influence of the Ar/D ratio it is important that both the above-mentioned parameters are constant during deposition. For an easier comparison, an index was used for each of the four batches of Be-D layers. For example, the Be-D layers deposited at a D flow of 2 sccm, has the attributed index of D2. The parameters employed for the deposition of the Be-D layers are listed in Table 1 together with their respective index.

**Table 1.** The main deposition conditions of the studied Be-D layers.

| Sample Index | Input Power (W) | Ar Flow (sccm) | D Flow (sccm) | Deposition Rate (nm/s) |
|:---:|:---:|:---:|:---:|:---:|
| D2 | 175 | 20 | 2 | 0.06 |
| D4 | 175 | 20 | 4 | 0.06 |
| D10 | 175 | 20 | 10 | 0.06 |
| D20 | 175 | 20 | 20 | 0.06 |

*2.2. Layer Analysis*

SEM images were taken from the samples deposited on tungsten and silicon substrates to compare the morphology of the layers. A FEI Co. model Inspect S (Hillsboro, OR, USA) was used to perform this task with the following tuning specifications: working distance in the range 0–30 mm, high vacuum conditions ($5.6 \times 10^{-2}$ Pa), variable electron acceleration voltages between 0 and 30 kV.

Layer's crystalline structure was investigated using X-ray diffraction (XRD). The XRD experimental setup consisted of a Bruker D8 Advance diffractometer (Coventry, West Midlands, England), in Bragg–Bretano geometry. Taking into consideration the low thickness of the Be-D layers combined with the polycrystalline nature of the tungsten substrate (tungsten peaks could overlap with the peaks resulted from the layers), we chose to perform the measurements on layers deposited on single crystalline silicon substrates

(100). The measurement range was from 20 to 60 degrees using a step of 0.01 with a 4 s integration time per step.

The elemental composition of the Be-D layers was investigated using RBS. As RBS does not provide an accurate quantification for low Z elements like D and Be, the main purpose of this measurements was primarily to detect the impurities that are present in the layers. A high oxygen contamination can heavily influence both D inventory in the layers and also its desorption kinetics, making an accurate interpretation very difficult if not impossible. However, the oxygen contamination of Be layers is unavoidable even for experiments performed in ultrahigh vacuum conditions [22]. The Be-D layers deposited on graphite substrates were placed on a goniometric holder (Automatic 3-Axis Goniometer, Panmure Instruments Ltd., Newtown, UK) with step size precision of 0.01°. A solid state silicon detector (AMETEK type BU-012-050-100, AMETEK Inc., Berwin, PA, USA) with an energy resolution of 18 keV was positioned at an angle of 165 degrees in relation to the ion beam to detect the backscattered protons. To determine the elemental constituents of the measured Be-D layers from the experimental spectra a simulation software developed at IPP Garhing was used (SIMRA) [23].

Deuterium desorption from trapping sites as well as its inventory in co-deposited layers were investigated using Thermal Desorption Spectroscopy (TDS). An important factor that may affect deuterium release from the sample is the exposure to atmospheric air and humidity. Note that sample exposure under these conditions was around 3 h. The measurements were carried out for layers deposited on tungsten and silicon substrates. The experimental system comprises in a quadrupole mass spectrometer (Quadera QMG 220, Pfeifer Vacuum, Asslar, Germany), a quartz tube and a furnace (Carbolite Gero, Neuhausen, Germany). All samples are loaded into the quartz tube which is directly linked through a flange with the mass spectrometer and the vacuum system. The system is pumped down by a turbo molecular pump up to a base pressure of $10^{-6}$ Pa. The quartz tube is divided in two areas: the loading area where all investigated samples are stored and the heat treatment area where each individual sample is measured, respectively. Before the commence of the experiment the heat treatment area is outgassed by heating it to 1300 K. After this step each individual sample is transported in this area and heated by means of a temperature-controlled oven, adjusted to maintain a stable 10 K/min heating rate until the final programmed temperature of 1300 K. The release of the desorbed species as a function of temperature is monitored by the QMS. Prior to TDS measurements on the Be-D layers, calibrations were performed in order to determine the dependence between the ion current measured by the QMS and the flow of molecules from the heated sample. The quantification of mass 4 signal was done by introducing known amounts of $D_2$ gas into the TDS chamber through a mass flow controller. For mass 3 quantification, both $H_2$ and $D_2$ were leaked into the TDS chamber separately and the value of the calibration factor was calculated as an average of their corresponding signals.

## 3. Results and Discussion

### 3.1. Layers Morphology

Surface morphology is a very important parameter for the D retention studies. As it was showed by Dinca et al. [24] in a study concerning D implantation in Be and mixed Be-W layers, a textured layer morphology can provide a large surface area of implantation for the D ions, thus increasing the amount of D retained, especially if we take into consideration the low solubility of D gas in Be coupled with a small penetration (<50 nm) depth at ITER relevant energies. SEM investigations were carried out on D2, D10 and D20 respectively, to compare the influence of deuterium flow substrate nature on the Be-D layer morphology. The images taken from these samples are presented in Figure 2. Figure 2a shows the SEM image taken for the Be-D layer deposited at a deuterium flow of 2 sccm. The image illustrates a smooth morphology composed of small grains uniformly distributed along the sample surface. The same conclusions drawn from the D2 sample can also be extended

for the rest of the Be-D layers deposited on silicon substrates since there are no visible differences between the SEM images.

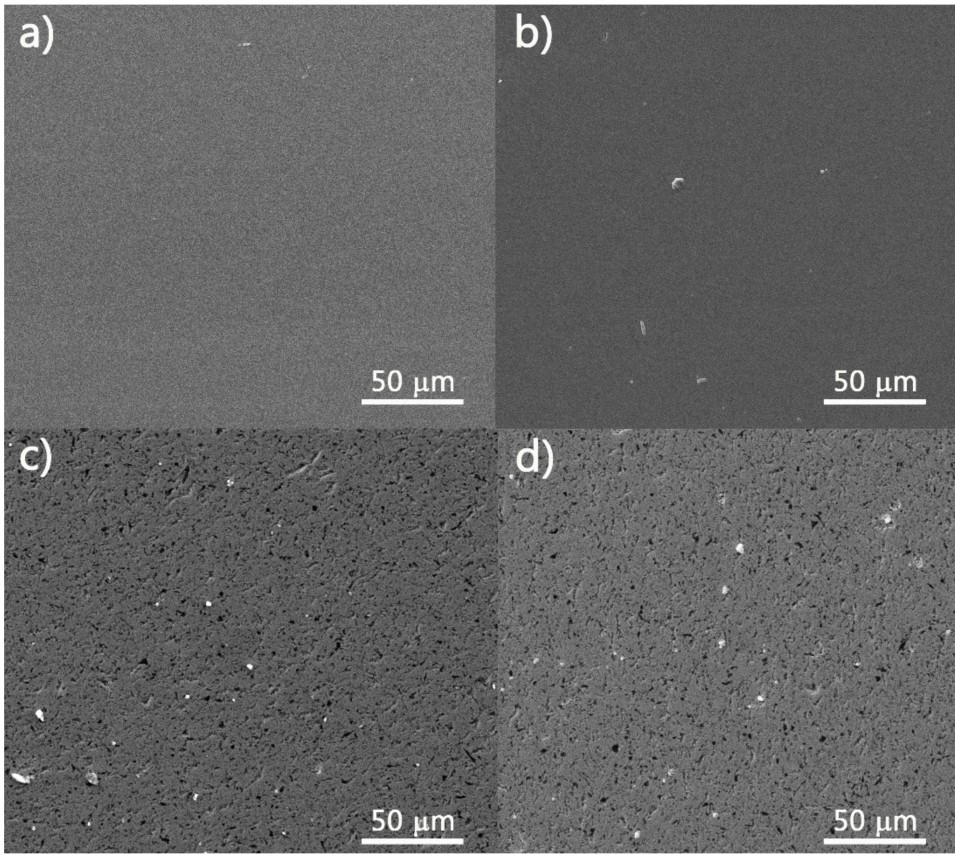

**Figure 2.** SEM images of the Be-D co-deposited layers: (**a**) D2 Si; (**b**) D20 Si; (**c**) D10 W; (**d**) D20 W.

On the other hand, the Be-D layers deposited on tungsten substrate present a different morphology altogether. The taken images point to a textured surface morphology. The voids observed on the surface could indicate a potential columnar growth of the layers. However, this particular hypothesis can be ruled out since there is no cross-section image for layers deposited on tungsten. One important parameter that can be responsible for the apparent differences between Be-D layers deposited on silicon and tungsten is the substrate roughness. In this case, the silicon substrate has low mean average roughness (6 nm) compared to W (50–70 nm). Substrate nature can have an important influence in promoting different growth mechanisms as it was showed by Jijun Yang et al. [25]. All the above can be concluded that in this case the surface morphology of the Be-D layers is mainly influenced by the substrate properties and not by the D flow during deposition. Another interesting observation for the D10 (Figure 2c) and D20 samples (Figure 2d) is the presence of micrometric particles on the surface of the layers. This can be evidence of the Be magnetron target transition from a "clean" sputtering regime to "poisoned" regime. This raises the possibility that beryllium deuteride is formed on the target surface, leading to the occurrence of arcing events that in the process expulse large particles.

### 3.2. Layers Crystalline Structure

XRD measurements were performed on three out of four Be-D layers deposited on silicon substrate: D2, D4 and D20. The diffraction patterns of the respective samples are illustrated below in Figure 3.

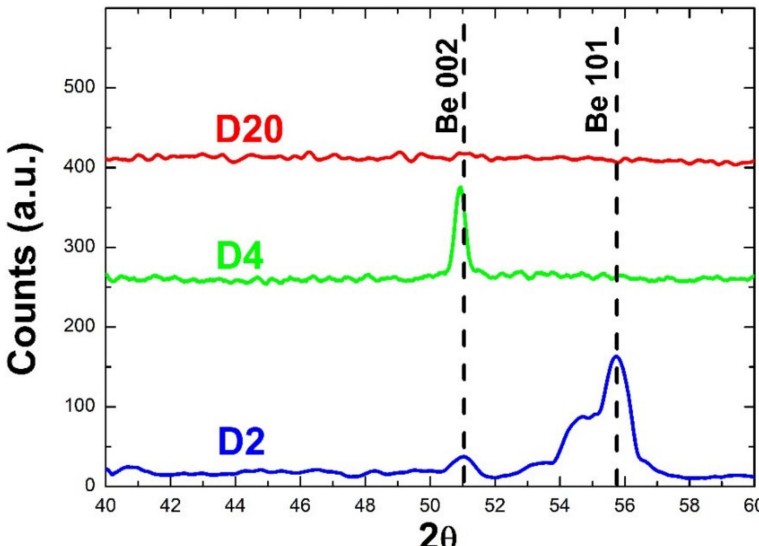

**Figure 3.** XRD diffraction patterns of the Be-D layers (D2, D4 and D20) deposited at 343 K on silicon substrates.

Opposite to the high integration time per step used in the measurements, the intensities of the Be peaks corresponding to different orientations are still very low due to the layers reduced thickness (500 nm) and beryllium transparency properties to X-ray. The diffraction pattern of the D2 sample indicates the presence of two peaks, the first one centered at a 2θ value of 51.14 degrees and the second at 53.14 degrees. Both peaks are corresponding to Be rt metallic phase, with a closed-packed atom arrangement in the unit cell, space group P6$_3$/mmc (194), with hkl orientations (002) and (101), respectively. The D2 sample crystallites exhibit strong preferential growth in (101) orientation. Furthermore, the (101) peak has a broad profile underlying the presence of small nanocrystallites in the layers.

In contrast to D2 sample polycrystalline texture, the diffraction pattern of the D4 sample presents only a single sharp peak corresponding to crystalline growth in (002) diffraction plane. For a deuterium flow of 20 sccm (D20) no diffraction peaks were observed, which is a clear indication of the amorphous nature of the layers. It underlines the fact that the increase of deuterium partial pressure during deposition can strongly influence the crystalline nature of the beryllium layers. It could also mean a significant build-up of deuterium gas in the sample.

The results show that the operating pressure range in ITER could significantly affect the crystalline structure and deuterium amount retained in Be-D layers formed in the cooler (373 K) areas of the inner vessel [26].

### 3.3. Layers Composition

RBS measurements were performed on all Be-D layers deposited on graphite substrate. The analysis carried out with this technique provides information regarding the thickness, chemical constituents, depth profile and composition of the layers.

In Figure 4 it is illustrated in comparison the D signal from the experimental spectra for the investigated samples. From the spectra it can be seen the signal corresponding to D is near 100th channel. It is noticeable that this signal becomes more intense with the increase of deuterium flow, with the lowest peak count for the D2 and the highest for D10 sample, respectively. This confirms that deuterium accumulates in the sample with the increase of deuterium flow during depositions. On the other hand, the Be signal (not shown here) decreases in intensity with deuterium flux increase. Thus, a higher deuterium flow introduced into the chamber during the co-deposition, leads to a lower sputtering efficiency of beryllium target.

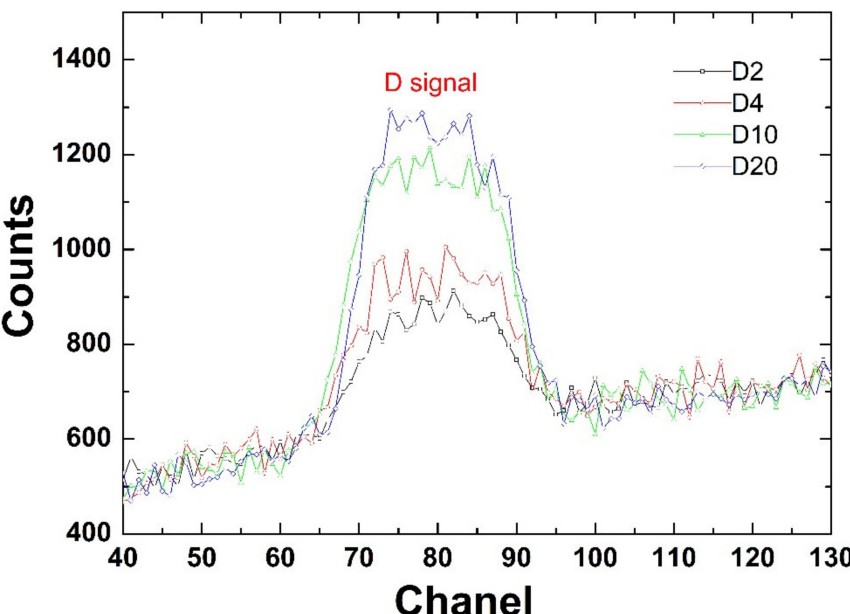

**Figure 4.** Deuterium signal extracted from the experimental RBS spectra measured for the Be-D layers deposited on graphite substrate.

The stoichiometric composition of the sample was evaluated using the areal densities resulted from the simulation of the experimental spectra. Furthermore, the thickness of the layers was determined by converting the areal densities into nanometers by approximating layer density to that of the bulk Be (1.803 g cm$^{-3}$). Both the compositional and thickness results for each individual sample are presented in the below table (Table 2).

**Table 2.** Chemical composition and thickness of Be-D layers resulted from the simulation of experimental RBS spectra.

| Sample Index | Be (at%) | O (at%) | D (at%) | Thickness (nm) |
|:---:|:---:|:---:|:---:|:---:|
| D2 | 85.36 | 2.43 | 12.19 | 581 |
| D4 | 84.70 | 2.35 | 12.95 | 598 |
| D10 | 75.00 | 2.27 | 22.72 | 548 |
| D20 | 71.40 | 2.61 | 26.19 | 498 |

In conclusion, as we expected from the first visual analysis of the experimental spectra the simulation results show both an increase of D and a decrease of thickness with the increase of D flow during deposition. Both results corroborate well the SEM and XRD results and confirmed that amorphous D10 and D20 sample also have the highest D inventory and that increased D flow reduces deposition rate due to target "poisoning". In principle two main mechanisms are responsible for the target "poisoning" phenomenon during reactive magnetron sputtering: implantation of the reactive species into the subsurface layers of the target and chemisorption represented by the adsorption of the reactive gas molecules on the target surface. Considering the low ion energies in a typical DC magnetron discharge, coupled with a small D ion implantation range in beryllium, chemisorption could be the main mechanism responsible for BeD compound formation on target surface. In this case, this effect should be more pronounced with the increase of deuterium flow leading to a decline of the deposition rates.

The D values obtained with RBS are between 100 and 300% higher than the ones resulted from the TDS measurements, and since NRA is presented in literature as the best suited method to determine the D inventory in the samples, we considered that in this particular case the information obtained by TDS is more reliable.

### 3.4. Deuterium Desorption and Thermal Release

Deuterium content quantification for the Be-D layers was calculated based on contribution from the HD (mass 3) and $D_2$ (mass 4) ion currents measured by the QMS. In a previous study by Dinca et al. [24] performed on Be-based samples implanted with deuterium, the D inventory was released predominantly in HD form. Opposite from that study, the results for Be-D co-deposited layers show that D is predominantly released in $D_2$ form. In this case, the $D_2$ ion current measured by the QMS was one order of magnitude higher compared to HD. As in the previous study, the release of D through HDO (mass 19) and $D_2O$ (mass 20) was carefully monitored but their respective ion currents were under the QMS detection limit. Figure 5 presents in comparison the desorption charts of the D2, D4, D10 and D 20 samples deposited on silicon and tungsten substrates. The resulted desorption curves were obtained by summing the contribution of HD and $D_2$ signals measured by QMS and normalizing the desorption to the unit area. Thereby the charts are expressed as $D/cm^2 \cdot s$ as a function of sample temperature.

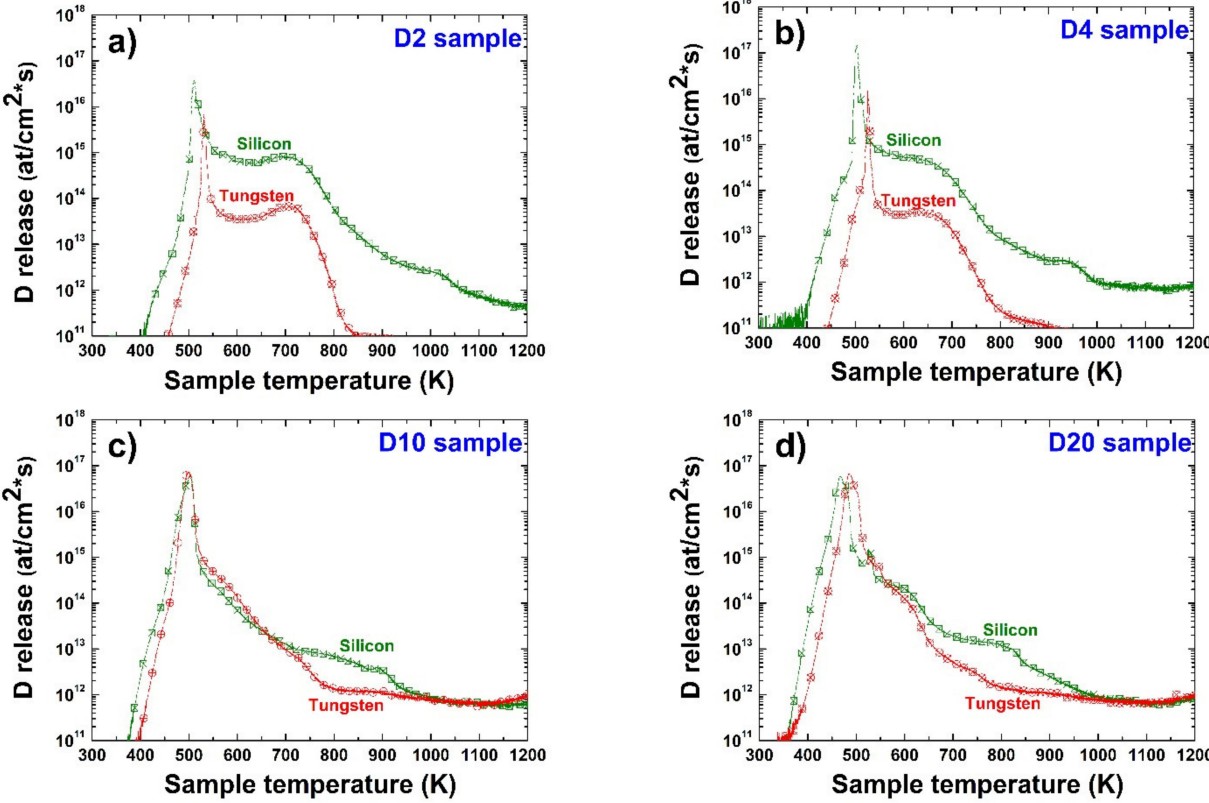

**Figure 5.** Deuterium release comparison for the Be-D layers deposited on tungsten and silicon substrates: (**a**) D2; (**b**) D4; (**c**) D10; (**d**) D20.

For the Be-D sample deposited at 2 sccm (D2) on silicon substrate (Figure 5a), the desorption curves indicate the presence of an intense sharp peak with a maximum intensity at 510 K and a main release between 400 and 550 K. In the literature, this peak was observed in several papers and particularly discussed by Markinn in et al. [27] and Reinalt et al. [28]. In these studies, the D release observed in the 450–500 K temperature range is usually associated with D trapping in supersaturated Be-D sites in the implanted region. This occurs only at high D fluency when all energetically favorable binding states are occupied, leading to D trapping in low activation unfavorable sites. Another feature observed is a small desorption shoulder with a maximum release intensity ~580 K. This peak was also observed by Doerner et al. [29] and is in good agreement with studies performed with the exposure of Be samples to deuterium plasma and also with those carried out on Be-D co-deposited samples by Temmermann et al. [21]. D2 sample desorption chart also exhibits

a large release feature similar to a shoulder between 630 and 730 K, with a maximum desorption at 700 K. This release behavior was also observed by Reinalt et al. [28] and is associated with D trapping in Be-O binding sites.

Figure 5b presents the desorption chart of the D4sample co-deposited on silicon substrate. Its D thermal release profile is dominated by an intense sharp peak observed at ~500 K similar with the one observed for the D2 sample, followed by a broad shoulder with a desorption onset at 630 K. However, this release feature is less prominent compared to the one observed from the D2sample. A small release of deuterium is observed at ~940 K. In accordance with Reinalt et al. [28], this peak is associated to deuterium desorption from high energy traps which are energetically favorable for D binding. These high energy traps can be considered as defects in the crystalline structure of the layers. The defects are created during the deposition by ion bombardment with energetic Be, Ar and D ions. Usually in studies conducted with D implantation this type of defect is created by the D ions impinging into the Be crystalline structure, and their formation being mostly dependent on the ion implantation energy and fluency. At small fluency values this trapping mechanism represents the dominant component of the D release chart, being characterized by broad desorption peaks at temperatures above 700 K. As it can be seen in the current study the D retention in defects is negligible compared to the rest of the observed retention mechanisms.

For the D10 sample deposited on silicon substrate (Figure 5c), the D desorption has an early onset at 370 K and continues up to the final measured temperature. The desorption chart points to a single dominant desorption feature with its peak value at 500 K and a small release feature at 900 K. Compared to the D4 and D2 samples presented earlier the D10 sample does not show BeOD specific release feature (630–700 K) even thaw the O content measured by RBS is similar for all samples. On the other hand, the D20 sample (Figure 5d) points to a textured desorption chart from a D trapping perspective. Similar with the rest of the samples the D release chart is dominated by a large peak at 476 K followed by a small sharp peak at 532 K. This release behavior was observed by Markin et al. [27] at high D irradiation fluency. Furthermore, the desorption shoulder specific to beryllium oxide trapping sites release starts at 630 K, and the release specific to high energy trapping sites is observed 770–840 K.

In conclusion, all Be-D samples deposited on silicon substrates present similar D thermal release behavior, and, implicitly, the same trapping mechanisms govern the D retention in these types of layers. However, after a careful analysis, several differences can be observed especially between the samples deposited at high D flow (D10, D20) compared to the ones deposited at low D flow (D2, D4). The first difference is connected to desorption onset, which begins at lower temperature (350 K and 370 K) for the D10 and D20 samples compared to higher than 400 K for the D2 and D4, respectively. Another difference can be observed in the shifting of the low energy activation peaks toward lower temperatures (approximately a 50 K shift for D20 compared to D2) with the increase of D flows. Furthermore, this was accompanied by a transition from a narrow-sharp peak profile towards a wider profile which can point out to a build-up of deuterium in these energetically unfavorable binding sites. The broadening of the D release peak coupled with the lower temperature shift observed in the desorption chart could also indicate the presence of an additional low energy binding sites. The occurrence of this binding sites is in close relation with the increase of the irradiation fluency, as it was observed by Reinalt et al. [28]. In this case it can be considered that increasing the deuterium flow during deposition can have a similar effect on the release behavior with the increase of fluency during D implantation. Another hypothesis that must be considered is the desorption peaks tendence to shift towards lower temperatures with the decrease of the layer thickness. This phenomenon was observed in three separate studies [30–32]. This hypothesis is supported by the RBS results, which confirm a decrease of the layer thickness with the increase of the deuterium flow during deposition.

The desorption charts of the Be-D layers deposited on tungsten substrate are also illustrated in Figure 5. They show similar behavior in terms of D release and trapping

mechanisms with the Be-D layers deposited on silicon substrates. In a similar fashion, the desorption charts are characterized by a massive release of deuterium between 470 and 520 K. The D release from D2 sample present a sharp low temperature peak at 520 K, followed by a broad release component between 630–730 K. Compared to D2 sample deposited on silicon, in this instance the desorption begins at a higher temperature (450 K compared to 400 K) and drops below the detection level of the QMS at 850 K unlike the first for which the desorption continues up to the final programmed temperature. The D4 sample also illustrates the same two-stage D desorption behavior as D2 sample, with the exception that the broad release shoulder appears at a lower temperature (630–681 K). Both D2 and D4 samples deposited on tungsten substrate do not have any desorption peaks associated with D release from high energy traps as it was the case with their counterparts deposited on silicon substrate. The D10 and D20 samples show broad desorption peaks at 500 K and 486 K, respectively. Additionally, the D20 sample has a small narrow desorption peak at 545 K. In terms of D release behavior and trapping mechanisms we can conclude that D10 and D20 samples deposited on tungsten are nearly identical with their counterparts deposited on silicon.

In an early paper published by the main authors of this study [24], concerning the D retention in pure and mixed Be-W layers, it was shown that the largest part of the deuterium inventory is released at temperatures higher than 700 K. It is important to mention that in the study a deuterium plasma torch was used to perform D implantation in Be layers. The main trapping mechanisms were BeOD and high energy binding states (defects) created during layer deposition and subsequent D irradiation. In contrast the D release behavior from Be-D co-deposited layers is very different. In this instance, D is mainly trapped in low energy binding states being released at temperatures below 700 K (99.99%).

D inventory was calculated by integrating the desorption curves as a function of time the total. Based on the RBS elemental densities for Be, the D retention is expressed in Figure 6 using atomic %. The retention for Be-D layers deposited on tungsten and silicon substrates is also compared in Figure 6. The lines that connect the points in the graph are for guiding purpose only. The results with the exception of the D4 sample are similar for both type of substrates indicating that both the substrate nature and the different layer morphologies do not have a strong influence on the total deuterium retention. Moreover, the D retention in D20 samples for different substrates is nearly identical. A parameter that clearly influences the retention is the D flow. A quasi-linear increase of retention with D flow during deposition can be observed. Furthermore, a 300% increase of D retained is observed for the D20 sample in comparison to D2.

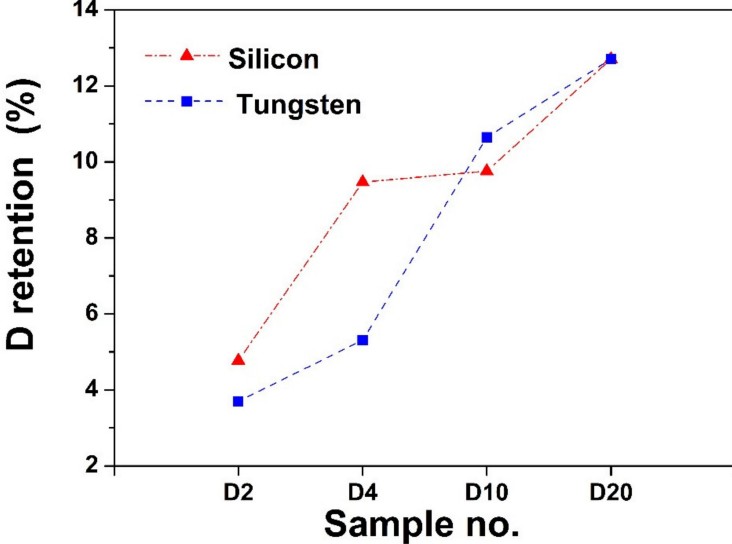

**Figure 6.** Deuterium retention comparison for the Be-D layers deposited on tungsten and silicon substrates.

## 4. Conclusions

In this manuscript, the influence of Ar:D ratio on D retention and thermal release behavior from Be co-deposited layers was investigated. Several conclusions have been drawn from this study and the most important are presented below. Results show that changes in of D flow do not induce morphological changes; however, the main promoter for different morphology observed between Be-D layers deposited on tungsten and silicon could be the substrate roughness. Gradual increase of D gas flow during deposition leads to the growth of amorphous Be layers. RBS results indicate that samples accumulate deuterium with the increase of D flow. Layer thickness shows a reverse dependence in respect to D flow rise, due to the Be target poisoning during deposition. The D retention in all Be-D co-deposited layers is mainly characterized by the D trapping in low energy activation binding states. The thermal release behavior for the majority of samples is dominated by a main sharp peak (450–500 K). The increase of the D flow leads to the widening of the main release peak and adds an additional low temperature trapping state similar to the one observed in implantation experiments performed at high fluency. In contrast to Be implanted samples, the majority of the D inventory in Be-D co-deposited layers is released at temperatures lower than 700 K. This indicates that D removal procedure applied in ITER (heating the divertor and walls to 623 K) has the potential to be very efficient for the Be-D co-deposited layers. The measured retention in samples points to a quasi-linear growth with the D flow, showing a 300% increase for D20 sample compared with D2.

**Author Contributions:** Conceptualization, P.D. and C.P.; Methodology, P.D. and C.S.; Validation C.P.L. and C.P.; Investigation, O.G.P., A.-M.B., C.S., B.B., I.B. and P.D.; Data curation, P.D.; Writing—original draft preparation, P.D.; Writing—review and editing, C.P.; Visualization, P.D. and B.B.; Supervision, C.P. and C.P.L.; Project administration; P.D.; Funding acquisition, P.D. All authors have read and agreed to the published version of the manuscript.

**Funding:** This work was supported by a grant of the Romanian Ministry of Education and research, CNCS-UEFISCDI, project number PN-III-P1-1.1-PD-2019-1024, within PNCDI III, and also received funding within the framework of the EUROfusion Consortium and has received funding from the Euratom research and training programme 2014–2018 and 2019–2020 under grant agreement No 633053. The views and opinions expressed herein do not necessarily reflect those of the European Commission.

**Institutional Review Board Statement:** Not applicable.

**Informed Consent Statement:** Not applicable.

**Conflicts of Interest:** The authors declare no conflict of interest.

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
