# Peer review of "Deuterium Retention and Release Behavior from Beryllium Co-Deposited Layers at Distinct Ar/D Ratio"

_coatings, doi:10.3390/coatings11121443_

Round 1

Reviewer 1 Report

The research paper entitled "Deuterium retention in beryllium co-deposited layers" contains some interesting findings.

But major revision is required before publication.

The authors have to address the folowing points:

  • Title should change with novelty.
  • What is SEM in Abstract.
  • Introduction should separate the paragraph to follow easily.
    The novelty need emphasize in last part in Introduction.
  • The author should refer some papers: Langmuir 37 (9), 2963-2973 (2021); Microchemical Journal, 106481 (2021
  • It is better to conduct AFM to evaluate layer analysis
  • To determine Deuterium desorption, it should quantify Deuterium 
  • SEM image is not clear.
  • Many mistakes in English wring must be corrected. 
    The author must correct "et al." not "et all"
  • Discussion on the results are weak.
  • Conclusions should again emphasize new findings in the present study.

Reviewer 2 Report

This article reports studies of determining the amount of deuterium retention in beryllium co-deposited layers by DC magnetron sputtering process on three types of substrates namely tungsten, silicon and graphite. Of course, this analytical article will show some interests to the research communities.

There are few questions regarding this article:

  1. The deposition rate keeps constant at 0.06 nm/s (see Table 1) in various deuterium flow of the sputtering process. This 0.06 nm/s rate value is a sputtering rate based on the measurement result from a quartz microbalance (see Line 236) not a deposition rate. It needs to clarify.
  2. In addition, in Table 2. the RBS results show the deposition thickness of Be-D layers was decreasing with increase of deuterium flow, means it is not a constant deposition rate throughout the D2-D20 experiments although it was explained by the reason of the sputtering target “poisoning” effect. It needs to elaborate more.
  3. If the input power keeps constant at 175 W in Table 1, why your results showed current density varies with 0.25-0.3 mA/cm2 (Line 230)( for this case, the plasma density will have a variation )? This variation is due to a small variation of input power during the experiment? It will result in the variation of deposition rate and will not have a constant deposition rate for the studies? Need to provide more clarity in the statement of the text.
  4. Line 242, …. the Be-D layers deposited at a D flow of 2 sccm. It should be at a D2 flow, instead of at a D flow, missing 2 in front of D.

Reviewer 3 Report

The manuscript is well organized and the results are clearly described and commented.

I can suggest to acceopt with minor revision. at line 499 they mention fig 7, but this does not exist. please correct

Reviewer 4 Report

I have not too many important remarks. In other words, I assume this manuscript is well-elaborated, the research novel on a world scale, the experimental methods suitable, the results presented in a proper manner, and their discussion at an appreciated scientific level. Nevertheless, some points need to be improved.

I have no comments on the title and the abstract. 

As concerns the Introduction:

  1. Line 128: In order solve these problems; in order to solve …
  2. Line 139-141: which also represents a test platform for the development of International Thermonuclear Experimental Reactor (ITER), which represents; twice the word “represents” is strange, please change it.
  3. Line 157: 700 g; what is it?

As regards the materials and methods, it is obligatory to show the details (a name of a tool, a company, a town, and/or a country) of either delivering company or a manufacturer for:

  1. (line 200) turbo molecular pump and a dry scroll pump
  2. substrates (Be, D, W, Si, graphite, argon) and their purity and form
  3. (line 219-220) mass flow controller (MFC)
  4. quartz microbalance
  5. ultrasonic cleaning device
  6. diffractometer
  7. Rutherford backscattering spectrometry (EBSD?)
  8. goniometric holder
  9. solid state silicon detector
  10. Thermal Desorption Spectroscopy

Here and in the next chapters some grammar or technical errors appear as:

  1. A lack of commas before “namely” (line 204) or “respectively” (lines 213, 355, 450) or “the intensity” (line 330).
  2. The improper grammar form: gas is feed; gas is fed.
  3. current density varies; current density changed.
  4. The technical error: coatings were performed; coatings were obtained (or: deposited).
  5. There is no space: 1Pa (line 215); 1 Pa.
  6. c.c.m. (Table 1); sccm.
  7. Numbered figures and tables should start with capital letters (lines 243, 298, 356, 470, 498, 499).
  8. The symbols (circles) of grades (degrees) seem inappropriate, in particular their positions (lines 257, 258, 334).
  9. Line 261: like D and Be the main purpose; unclear.
  10. Line 282: were all investigated samples are stored; where all …
  11. Line 284: baking it to 1300 K; baking?

The results and discussion, remarks:

  1. Repeatedly, many times, the authors improperly write “et all” instead of et al.”. For example, limes 314, 381, 401, 409, 411, 421, 459.
  2. Line 295: at ITER relevant energies SEM investigations; unclear, a lack of a point?
  3. Line 312: the authors assume “In this case the silicon substrate has low mean average roughness (6 nm) compared to W (50-70 nm).” I cannot see such results, please prove and show them, e.g., by AFM results.
  4. Line 332: X ray; X-ray.
  5. Line 333: pattern of the D2 sample indicate; … indicates.
  6. Line 335: Be rt metallic phase closed –packed structure; unclear, what is Be rt? The phase of close-packed?
  7. Line 337: growth on (101) orientation; growth in?
  8. Line 340: pattern of the D4 sample present; … presents.
  9. Line 345: however these could also mean; however, it could …
  10. Line 354: to determine their respective composition; … compositions.
  11. Line 356: In figure 4 is; In Figure 4 it is.
  12. Lines 360-361: This confirms that deuterium accumulates in the sample in relation deuterium flow increase during depositions; unclear, in relation to?
  13. Line 386: (mass4; (mass 4.
  14. Lines 389-390: release from the current B-D co-deposited layers was dominated by D2 which is one order of magnitude higher than HD; unclear.
  15. Line 402: the D release observe; … observed.
  16. Line 403: D trapping in supersaturated Be-D states in the implanted region; the “state” should be substituted by “site” (see also lines 412, 457).
  17. Line 409: desorption chart also exhibit; … exhibits.
  18. Line 425: this type of defects are created; … is created.
  19. Line 432: desorption chart point to; … points to.
  20. Line 438: D release chart is dominate; … is dominated.
  21. Line 445: an implicitly the same trapping; and implicitly …
  22. Lines 459-461: increasing the deuterium flow during deposition can have a similar effect on the release behaviour with the increase of fluency during D implantation; unclear.
  23. Line 473: desorption spectra of the D2 sample presents; … present.
  24. Lines 475-476: begins at a higher temperature and (450 K compared to 400 K) and; improper phrase.
  25. Line 477: unlike the silicon were the desorption continues up; unlike the silicon for which …
  26. Line 480-481: samples deposited on tungsten substrate don’t have a desorption peak; have any desorption peaks.

I have no remarks on the Conclusions even if they appear too big; the only most important conclusions drawn from the results should be given. Please think about it.

All remarks are indicated in color in the attached reviewed manuscript.

Round 2

Reviewer 1 Report

The revised paper is suitable for publication in Polymers.